# Factors Predicting 30-Day Grade IIIa–V Clavien–Dindo Classification Complications and Delayed Chemotherapy Initiation after Cytoreductive Surgery for Advanced-Stage Ovarian Cancer: A Prospective Cohort Study

**DOI:** 10.3390/cancers14174181

**Published:** 2022-08-29

**Authors:** Malika Kengsakul, Gatske M. Nieuwenhuyzen-de Boer, Suwasin Udomkarnjananun, Stephen J. Kerr, Helena C. van Doorn, Heleen J. van Beekhuizen

**Affiliations:** 1Department of Gynecologic Oncology, Erasmus MC Cancer Institute, University Medical Center Rotterdam, 3000 CA Rotterdam, The Netherlands; 2Department of Obstetrics and Gynecology, Panyananthaphikkhu Chonprathan Medical Center, Srinakharinwirot University, Nonthaburi 11120, Thailand; 3Department of Obstetrics and Gynecology, Albert Schweitzer Hospital, 3318 AT Dordrecht, The Netherlands; 4Division of Nephrology, Department of Medicine, Faculty of Medicine, King Chulalongkorn Memorial Hospital, Chulalongkorn University, Bangkok 10330, Thailand; 5Biostatistics Excellence Centre, Faculty of Medicine, Chulalongkorn University, Bangkok 10330, Thailand

**Keywords:** epithelial ovarian cancer, morbidity, postoperative complications, Clavien–Dindo classification, time to chemotherapy

## Abstract

**Simple Summary:**

Both surgical outcome and timely initiation of chemotherapy are essential endpoints after cytoreductive surgery for advanced-stage epithelial ovarian cancer (AEOC). This was a multicenter prospective study of 300 primary AEOC patients who underwent cytoreductive surgery. We aimed to evaluate factors associated with 30-day severe postoperative complication as according to Clavien–Dindo classification (CDC) grade ≥IIIa and delayed initiation of chemotherapy defined as time to chemotherapy (TTC) >42 days after cytoreductive surgery for primary AEOC. The understanding of these risk factors and their consequences offers an opportunity to improve future perioperative care for AEOC. Our study highlights that patient with CDC grade ≥IIIa had a significant longer median TTC compared to patients without CDC grade ≥IIIa. Intraoperative upper-abdominal visceral injury was the strongest factor associated with both CDC grade ≥IIIa and TTC >42 days. In our analysis, patient performance status was the only preoperative modifiable risk factor for TTC >42 days.

**Abstract:**

**Objective**: The aim of this study was to evaluate factors associated with 30-day postoperative Clavien–Dindo classification (CDC) grade IIIa or greater complications and delayed initiation of chemotherapy after cytoreductive surgery (CRS) for primary advanced-stage epithelial ovarian cancer (AEOC). **Methods:** This was a prospective study involving 300 patients who underwent primary or interval CRS for AEOC between February 2018 and September 2020. Postoperative complications were graded according to the CDC. Logistic regression analysis was used to evaluate factors predicting CDC grade ≥IIIa and time to chemotherapy (TTC) >42 days. **Results:** Interval CRS was performed in 255 (85%) patients. CDC grade ≥IIIa occurred in 51 (17%) patients. In multivariable analysis, age (*p* = 0.036), cardiovascular comorbidity (*p* < 0.001), diaphragmatic surgery (*p* < 0.001), intraoperative urinary tract injury (*p* = 0.017), and upper-abdominal visceral injury (e.g., pancreas, stomach, liver, or spleen) (*p* = 0.012) were associated with CDC grade ≥IIIa. In 26% of cases, TTC was >42 days (median (IQR) 39 (29–50) days) in patients with CDC grade ≥IIIa versus 33 (25–41) days in patients without CDC grade ≥ IIIa (*p* = 0.008). The adjusted odds ratio of developing TTC >42 days was significantly higher in patients associated with WHO performance grade ≥2 (*p* = 0.045), intraoperative bowel injury (*p* = 0.043), upper-abdominal visceral injury (*p* = 0.008), and postoperative CDC grade ≥IIIa (*p* = 0.032). **Conclusions**: Patients with advanced age, with cardiovascular comorbidity, and who required diaphragmatic surgery had an increased adjusted odds ratio of developing CDC grade ≥IIIa complications. CDC grade ≥IIIa complications were independently associated with TTC >42 days. Proper patient selection and prevention of intraoperative injury are essential in order to prevent postoperative complications and delayed initiation of chemotherapy.

## 1. Introduction

The standard treatment for advanced-stage epithelial ovarian cancer (AEOC) is a combination of primary cytoreductive surgery (PCS) and platinum-based chemotherapy, with or without maintenance therapy [1]. In addition, patients with medically inoperable disease or profound disseminated disease may benefit from neoadjuvant chemotherapy (NACT) followed by interval cytoreductive surgery (ICS) [2,3]. Regardless of whether the treatment strategy is an upfront surgery or ICS, the degree of residual tumor following cytoreduction remains the strongest predictor for primary AEOC outcomes, and current best practice is to achieve complete cytoreduction (no visible tumor) [4,5].

To achieve this aim, surgery may require pelvic and/or abdominal peritonectomy, bowel resection, splenectomy, partial hepatectomy, and diaphragmatic stripping in cases of advanced-stage disease. Such extensive surgery is strongly related to postoperative morbidity and mortality [6]. Severe morbidity increases the time to chemotherapy (TTC), and this adversely affects the patient’s prognosis [7,8,9]. Generally, it is recommended to start or resume chemotherapy as soon as possible after cytoreductive surgery. Several studies have shown that initiating adjuvant chemotherapy >42 days after cytoreduction is independently associated with adverse survival outcomes [10,11,12]. A recent study demonstrated that women with TTC ≤ 42 days had a significantly better median progression-free survival (PFS) compared to those with TTC > 42 days (35.5 vs. 22.6 months) [10]. A meta-analysis in patients with International Federation of Gynecology and Obstetrics (FIGO) stage III–IV showed that overall survival (OS) declined by 4% for each week that adjuvant chemotherapy was delayed [13]. As a result, rapid recovery after surgery and timely initiation of chemotherapy are essential for optimizing ovarian cancer treatment outcomes.

During the past decade, several strategies designed to improve surgical outcomes in patients with gynecological malignancies have been implemented [14]. An enhanced recovery after surgery (ERAS) protocol was developed and recommended in an international guideline for management of gynecological malignancy [15]. Key features of the ERAS protocol include comprehensive preoperative counseling and risk modification, decreased preoperative fasting time, avoidance of bowel preparation, standardized analgesic and anesthetic regimens, and early enteral feeding and mobilization [16,17]. The application of the ERAS protocol in cytoreductive surgery has led to a reduction in postoperative recovery time, length of hospital stay, and TTC [18].

To monitor the quality of surgical care, a standardized system is necessary to classify surgical complications. The Clavien–Dindo classification (CDC) was first described in 2004 [19], and it became widely accepted in both general and urological surgery because of its validity and consistent interpretation [20]. The systematic evaluation of complications after maximal cytoreduction for ovarian cancer in the ERAS protocol is still limited. In fact, there is no consensus on how to classify complications and outcomes following surgery on gynecological malignancies. Our primary aim was to analyze factors associated with severe 30-day postoperative morbidity, defined as CDC grade ≥IIIa, in patients who underwent either primary or interval cytoreductive surgery for AEOC. Our secondary aim was to evaluate factors associated with a delay in initiating chemotherapy, defined as TTC >42 days after cytoreductive surgery. Identification of these potential risk factors and their consequences offers an opportunity to improve perioperative care in the future.

## 2. Methods

### 2.1. Study Design and Participants

This was a post hoc analysis of data from the “PlasmaJet surgical device in the treatment of advanced stage ovarian cancer (PlaComOv)” study, a prospective multicenter single-blinded randomized controlled trial conducted in nine teaching and four university hospitals in the Netherlands from February 2018 to September 2020 [21]. In the Netherlands, patients with suspected ovarian cancer have been cared for in centralized cancer centers since 2012 [22]. All patients are discussed at multidisciplinary tumor board meetings. In the PlaComOv Study, patients with suspected AEOC, fallopian tube, or peritoneal carcinoma FIGO stage IIIB–IV who were suitable for cytoreductive surgery and chemotherapy were eligible for inclusion. NACT consisted of three cycles of three-weekly intravenous paclitaxel (175 mg/m^2^ body surface area) combined with carboplatin (area under the curve of 6 mg mL/min) prescribed to patients who were not eligible for optimal cytoreductive surgery (i.e., unresectable tumors based on imaging, stage IV disease or WHO performance grade ≥3) followed by ICS. Briefly, patients were randomized to standard care plus use of the PlasmaJet surgical device during cytoreductive surgery or to standard of care only. PlasmaJet training and certification were provided to all attending surgeons [21]. Perioperative management was conducted according to the ERAS protocol [16] except in those who received hyperthermic intraperitoneal chemotherapy (HIPEC) (Appendix A).

In this current study, patients with recurrence of disease, nonepithelial or borderline ovarian tumors, or nonovarian malignancy, as well as patients who did not undergo surgery after randomization due to deterioration in performance status or progressive disease, were excluded. Patients in whom surgery was abandoned because of unresectable disease were also excluded (Figure 1). All patients gave written informed consent for the PlaComOv study, which was approved by the Medical Ethical Committee of the Erasmus Medical Center Rotterdam (number: NL62035.078.17 and trial registration number NTR6624 (registered 18 August 2017)).

### 2.2. Variables and Definitions

Patient demographics and study-related variables included age at surgery, body mass index (BMI), World Health Organization (WHO) performance status, smoking status, comorbidities, surgical history, randomized group (with/without PlasmaJet device), and type of surgery (PCS or ICS). Perioperative variables included the presence of ascites, tumor dissemination pattern, surgical procedures, surgical complexity (standard or extensive), intraoperative complications, estimated blood loss, operative time, surgical outcome, and HIPEC procedure. Postoperative variables included intensive care unit (ICU) admission, re-exploration indications, 30-day postoperative complications, readmission indications, time to adjuvant chemotherapy, tumor histology, and FIGO stage. All patient clinical characteristics and operative reports were retrieved from an electronic database in which they had been prospectively recorded. Extensive surgery was defined as any of the following procedures: peritonectomy, diaphragmatic peritonectomy, resection of subcapsular liver metastases, splenectomy, bowel resection, or resection of extra-abdominal metastatic sites. Intraoperative injury was defined as an undesirable and unintended result of an operation affecting the patients occurring as a direct result of the operation [23]. Intraoperative urinary tract injury was defined as any injury of bladder, ureter, or kidney. Intraoperative bowel injury was defined as any injury of the small bowel or large bowel. Upper-abdominal visceral injury was defined as any injury of the pancreas, stomach, liver, or spleen. All injuries were reported by the attending gynecologic oncologist present during the procedure. Each patient received daily postoperative visits until discharge. Any deviations from the normal postoperative course were recorded and graded according to the Clavien–Dindo classification system (Appendix A) [19]. If the patients had more than one complication, only the most severe complication was included in the analysis. Patients were scheduled for outpatient clinic visits 2 weeks after surgery, and telephone follow-up was scheduled at 4 and 6 weeks.

### 2.3. Study Outcomes

The primary outcome was CDC grade ≥IIIa (severe complications), defined as any life-threatening condition or complication leading to invasive radiological intervention, re-exploration, ICU admission, single- or multi-organ failure, or death within 30 days of surgery [19]. The secondary outcome was time to initiate chemotherapy, defined as the period from the date of cytoreductive surgery to the beginning or resumption of chemotherapy. A delay in initiating chemotherapy was considered when time to initiate chemotherapy exceeded 42 days following surgery, or when the patients did not receive chemotherapy [24]. Patients who refused chemotherapy or did not receive chemotherapy due to lack of tumor response to neoadjuvant chemotherapy (NACT) or who were lost to follow up were excluded.

### 2.4. Statistical Analysis

Categorical variables are presented as numbers (%) and continuous variables are presented as means ± standard deviation (SD) or median (interquartile range, IQR) as appropriate. The numbers of intra- and postoperative complications are presented as numbers (%). Logistic regression analysis was performed to obtain unadjusted odds ratios (ORs) and adjusted odds ratios (aORs) with corresponding 95% confidence intervals (95% CI). The linearity of continuous variables against the logit function was assessed, and, in the case of nonlinearity, the variables were modeled in quartiles; adjacent quartiles were collapsed if the OR and 95% CI were similar. All variables with *p*-values <0.10 in univariable analysis were adjusted for in the multivariable model [25]. Statistical significance was considered for *p*-values <0.05. The analyses were performed with IBM SPSS statistics for Windows, version 22.0 (Armonk, NY, USA: IBM Corp) and Stata Statistical Software Release 17.0 (StataCorp LLC, College Station, TX, USA).

## 3. Results

### 3.1. Patient Characteristics and Surgical Outcome

Of the 327 patients randomized in the PlaComOv study, 300 met the eligibility criteria for our analysis (Figure 1). The mean patient age was 65 (SD 10.4) years. The majority of patients had good performance status; 175 (58%), 105 (35%), and 20 (7%) patients had WHO performance status grade 0, 1, and ≥2, respectively. Almost 96% (*n* = 287) of the patients were diagnosed with serous carcinoma. Patient comorbidities and baseline characteristics are presented in Table 1. NACT was given to 255 (85%) patients. Complete cytoreduction was achieved in 235 (78%) patients. The majority of patients, 209 (70%), required extensive surgical intervention. The type and frequency of surgical procedures are reported in Appendix A. Approximately 20% (*n* = 61) of patients underwent an HIPEC procedure. The median (IQR) operative time was 210 (150–300) min, and median blood loss was 850 (400–1357) mL. Intraoperative complications occurred in 82 (27%) patients. The three most common intraoperative complications were blood loss ≥1000 mL (*n* = 125), bowel injury (*n* = 21), and urinary tract injury (*n* = 12). The type and frequency of intraoperative complications are reported in Appendix A.

Of 300 patients, 147 (49%) developed at least one postoperative complication (CDC grade I–V) within 30 days of surgery. There were 57 CDC grade ≥IIIa complications in 51 (17%) patients: 23 grade III, 33 grade IV, and one grade V. Ten (3%) patients underwent re-exploration for intestinal anastomosis leakage (*n* = 2), intraabdominal bleeding (*n* = 1), pancreatic leakage (*n* = 1), gastric perforation (*n* = 1), intraabdominal abscess (*n* = 1), and peritonitis (*n* = 4). One patient died after an uneventful recovery at home on the eighth day after surgery, and no autopsy was performed. The type and frequency of medical and surgical postoperative complications (CDC grade ≥IIIa) are reported in Table 2. The median (IQR) length of hospital stay was 7 (5–9) days. Of 30 (10%) patients readmitted to the hospital, three were diagnosed with CDC ≥IIIa complications, namely, splenic hemorrhage, wound dehiscence, and acute cholecystitis.

### 3.2. Regression Analysis of Factors Related to Clavien–Dindo Classification Grade ≥ IIIa

The univariable and multivariable analyses of factors associated with postoperative CDC grade ≥IIIa are presented in Table 3. In univariate analysis, *preoperative parameters* associated with CDC grade ≥IIIa at *p* < 0.10 were increasing age and cardiovascular comorbidity. Patients who underwent an appendectomy, colon surgery, mesenteric tumor resection, diaphragmatic surgery, or colostomy, or who had a longer operative time had greater odds of developing CDC grade ≥ IIIa. *Types of intraoperative injury* related to CDC grade ≥IIIa were urinary tract injury and upper-abdominal visceral injury. Other factors, namely, HIPEC, complete cytoreductive, and FIGO stage, showed no univariable association with CDC grade ≥IIIa. The use of the PlasmaJet device during surgery and PCS was associated with lower odds of developing CDC grade ≥IIIa.

In multivariable analysis, five factors were independently associated with CDC grade ≥IIIa: age (aORs 1.26, 95% CI 1.02–1.58 per 5-year increase, *p* = 0.036), cardiovascular disease (aORs 4.75, 95% CI 2.07–10.91, *p* < 0.001), diaphragmatic surgery (aORs 4.26, 95% CI 1.93–9.43, *p* < 0.001), intraoperative urinary tract injury (aORs 6.00, 95% CI 1.38–25.93, *p* = 0.017), and upper-abdominal visceral injury (aORs 9.07, 95%CI 1.61–51.08, *p* = 0.012) (Figure 2A).

### 3.3. Regression Analysis of Factors Related to Time to Adjuvant Chemotherapy >42 Days

Of 300 patients, 25 patients did not receive chemotherapy after surgery and were excluded from the analysis of delayed chemotherapy initiation (Figure 1). The majority (74%) of patients received adjuvant chemotherapy ≤42 days after surgery, whereas approximately 26% (72/275) of patients received chemotherapy >42 days due to poor performance status and/or prolonged recovery after complications. The median (IQR) TTC was 34.5 (27–44) days. The median TTC in patients with CDC grade ≥IIIa was 39 (29–50) days, while the median TTC in patients without CDC grade ≥IIIa was 33 (25–41) days, *p* = 0.008. The univariate and multivariable analyses of factors associated with TTC >42 days are presented in Table 4.

In univariable analysis, *preoperative parameters* related to TTC >42 days were WHO performance status grade ≥2 and presence of comorbidity. TTC >42 days was associated with the following *intraoperative factors*: HIPEC procedure, increased operative time. and intraoperative injury (urinary tract injury, bowel injury, upper-abdominal visceral injury, or pneumothorax). Postoperative CDC grade ≥IIIa complication was the only postoperative factor related to TTC >42 days.

In multivariable analysis, patients with WHO performance status grade ≥2 (aORs 2.51, 95% CI 1.02–6.15, *p* = 0.045), intraoperative bowel injury (aORs 2.98, 95% CI 1.04–8.56, *p* = 0.043), upper-abdominal visceral injury (aORs 10.26, 95% CI 1.85–57.13, *p* = 0.008), and postoperative CDC grade ≥IIIa complications (aORs 2.32, 95% CI 1.08–5.01, *p* = 0.032) had significantly higher adjusted odds of developing TTC >42 days (Figure 2B).

### 3.4. Interval Cytoreductive Surgery

In a sensitivity analysis of patients who received ICS (*n* = 255), four factors were independently related to CDC grade ≥IIIa: cardiovascular disease (aORs 4.01, 95% CI 1.66–9.68, *p* = 0.002), diaphragmatic surgery (aORs 3.90, 95% CI 1.74–8.70, *p* < 0.001), intraoperative urinary tract injury (aORs 6.90, 95% CI 1.50–31.69, *p* = 0.013), and upper-abdominal visceral injury (aORs 9.97, 95% CI 1.53–65.08, *p* = 0.016) (Appendix A).

Patients who experienced intraoperative bowel injury (aORs 3.27, 95% CI 1.03–10.37, *p* = 0.045), upper-abdominal visceral injury (aORs 17.59, 95% CI 1.87–165.36, *p* = 0.012), and postoperative CDC grade ≥IIIa complications (aORs 2.47, 95% CI 1.11–5.49, *p* = 0.027) had a significantly higher adjusted odds of developing TTC >42 days (Appendix A).

## 4. Discussion

In the PlaComOv study, adjuvant use of the PlasmaJet surgical device during AEOC surgery resulted in a higher proportion of complete cytoreduction, while surgical complication rates were similar for patients receiving standard care plus PlasmaJet versus standard care alone [21]. We conducted a post hoc analysis of the PlaComOv study to evaluate variables associated with postoperative complications and delayed chemotherapy initiation.

Systems to classify postoperative complications have recently been introduced in several surgical fields. The key systems are disease-based or rely on treatment-based scoring methods [20,26]. Our study used the Clavien–Dindo classification (CDC) to assess and rank postoperative complications in an objective and reproducible manner [19]. Increasing severity of CDC is associated with poor surgical outcomes such as prolonged length of hospital stays, ICU admission, and mortality [27,28]. Among studies that used the CDC, the prevalence of severe postoperative complications (CDC grade IIIa–V) after cytoreductive surgery for ovarian cancer ranged from 10% to 25% [28,29,30,31,32]. The incidence of CDC grade ≥IIIa was 17% in our prospective study, in line with other studies.

The presence of comorbidities and advanced age are well recognized risk factors for postoperative morbidity and mortality [33,34,35]. A recent meta-analysis demonstrated that patients with postoperative complications were significantly older than patients without complications. Moreover, patients aged ≥60 years had double the odds of developing postoperative complications compared to patients younger than 60 years old [36]. In our study, the risk of CDC grade ≥IIIa was significantly increased with advanced age in overall patients (PCS and ICS); however, this factor was no longer significant in a sensitivity analysis including only patients who received ICS. Nonetheless, the effect size and 95% CI were similar in magnitude and direction, suggesting a loss of statistical power following the exclusion of some patients in the sensitivity analysis. Additionally, we found that cardiovascular disease (myocardial infarction, stroke, and peripheral or vascular disease) was independently associated with CDC grade ≥IIIa.

Studies of extensive surgery for AEOC reported an increased rate of complete cytoreduction after upper-abdominal surgery, at the cost of complications [26,32]. Diaphragmatic surgery, hepatic resection, pancreatectomy, and biliary surgery were significantly associated with severe postoperative complications [28]. This is in line with our findings, since patients who underwent diaphragmatic resection had a significantly greater adjusted odds of developing CDC grade ≥IIIa compared to patients who did not require this procedure. Another study carried out by Llueca et al. demonstrated that ≥5 visceral resection, rectosigmoid resection, glissectomy (liver surgery), and pelvic peritonectomy were independent risk factors for major complications [37]. We found that bowel surgery with colostomy, appendectomy, and mesenteric tumor resection were potential risk factors for CDC grade ≥IIIa in univariable analysis; however, the independent association was lost after adjustment in our multivariable model.

In general, ovarian cancer surgery is extensive, and surgeons commonly encounter intraoperative massive bleeding, visceral organ injury, and prolonged operation times [34]. Although increased operative time showed a univariable association with CDC grade ≥IIIa, it was not independently associated with CDC grade ≥IIIa in an adjusted model. We found that intraoperative complications, specifically urinary tract injury and upper-abdominal visceral injury, were significantly associated with CDC grade ≥IIIa, which is consistent with the results of earlier retrospective studies [24,29,38].

The ERAS protocol is now globally accepted as a perioperative practice which improves patient outcomes. A recent randomized control trial on the ERAS protocol reported that patients allocated to the ERAS group had a decreased median length of hospital stay compared to those allocated to the controlled group (7 vs. 9 days), as well as a decreased readmission rate (6% vs. 20%) [39]. In addition, there were no differences in the incidence of intraoperative or postoperative complications, reoperation during primary stay, or mortality. Our patients were generally managed accordingly to the ERAS protocol, and our complication rate, length of hospital stay, and readmission rates were comparable to the previous study [39].

The optimal time to initiate or resume systemic chemotherapy after AEOC is still debated. Some studies defined delayed chemotherapy as >28 days after surgery [9,40]; our study used a cutoff duration of 42 days based on the Dutch cancer register study [12]. Despite increased time to adjuvant chemotherapy after surgery having a negative prognostic effect on patient survival, there is no clear international consensus defining a cutoff threshold where risk definitely increases [41]. Moreover, studies seldom report factors associated with a delay in initiating adjuvant chemotherapy for ovarian cancer. In our analysis, approximately 26% of patients initiated chemotherapy >42 days after surgery. A retrospective study by Castro et al. reported an incidence of TTC >42 days of 50.6%, almost double that of our study. The authors reported that hypertension, BMI ≥30 kg/m^2^, reoperation, and fever within 30 days after surgery were the independent factors related to TTC >42 days [24]. In another study, preoperative factors related to delayed TTC in PCS for AEOC were age >65 years, albumin <3.5 g/dL, and high age-adjusted Charlson Comorbidity Index [42]. In the present study, we found no association of age and BMI (modeled as continuous and in clinically relevant categories) with TTC >42 days. However, preoperative WHO performance status grade ≥2 was a preoperative risk factor significantly associated with TTC >42 days. Moreover, intraoperative upper-abdominal visceral injury showed the strongest association with both CDC grade ≥IIIa complications and TTC >42 days. The results in sensitivity analysis were similar to the entire cohort analysis, except that WHO performance status ≥2 did not meet criteria for inclusion in the multivariable sensitivity analysis, although the effect size and the 95% CI were consistent with an increase in risk in the univariable analysis, possibly due to the reduced power.

Among 2948 gynecological cancer surgeries, Iyer et al. found an overall intraoperative complication rate of 4.7%. The highest intraoperative complication rate was encountered during ovarian cancer surgery. Intraoperative hemorrhage was the most common complication followed by bladder injury and small bowel injury [23]. Our findings were similar to a previous study. To minimize the risk of postoperative complications, an international guideline on perioperative management recommends that patients undergoing debulking surgery for ovarian cancer should be preoperatively managed against expected risks [43]. The authors stated that, although prophylactic stents are not associated with a decreased risk of ureteric injury, they may be considered in patients with a high risk of injury such as previous urological operation and/or preexisting hydronephrosis [43]. However, we found no modifiable preoperative risk factors for CDC grade ≥IIIa in our study. Furthermore, most urinary tract injuries were bladder injuries, and only one patient had a ureter injury which was repaired during the same operation. In spite of this, there remains a lack of guidelines for the prevention of bladder injuries. At present, single mechanical bowel preparation is not routinely performed. Postoperative fasting and routinely applied protective stoma formation to prevent anastomosis leakage are not recommended. Foremost, gynecological oncologists must be familiar with the abdominal anatomy and variation, especially when performing an extensive surgery [43]. A multidisciplinary team approach should be taken when performing upper-abdominal surgery and/or when encountering organ injuries.

In case of preventing TTC >42 days, WHO performance status was the only adjustable preoperative risk factor. Recently, a pilot study with preoperative pre-habilitation showed favorable results in feasibility, shorter length of hospital stay, and TTC. However, there has been no study clearly demonstrating the effect of this intervention on postoperative complications and survival outcomes [44].

## 5. Strengths and Limitations

We performed a large prospective cohort study to evaluate the factors predicting both severe postoperative complications and time to adjuvant chemotherapy >42 days. The main strength of our study is that all parameters were prospectively and systematically collected from the PlaComOv randomized controlled trial. Patient characteristics, as well as intraoperative and postoperative information, were categorized and collected in a uniform manner in a monitored electronic database, which reduces the risk of missing or incorrect data. Secondly, all included hospitals were high-volume ovarian cancer surgery centers. Every case in our cohort was centralized to the registered centers. The surgeons applied maximal effort to remove all visible tumors, and several extensive procedures were performed. Although data from this study were from an RCT, we excluded patients who declined surgery following deterioration in performance status, patients with progressive disease, and patients in whom surgery was abandoned. In addition, some variables, such as preoperative frailty index, albumin level, intraoperative peritoneal carcinoma index, and ERAS compliance, which are known to influence outcome, were not available in the study database and, therefore, not included in our analysis. In an attempt to mitigate bias caused by these issues, we adjusted for all potential confounders in our multivariable model, but the possibility of unobserved confounding factors cannot be excluded.

## 6. Conclusions

In the era of complete cytoreductive surgery to optimize outcomes in AEOC patients, individual patient factors influence the balance between the benefits of extensive cytoreductive surgery and postoperative morbidity. Patients with advanced age, with cardiovascular comorbidity, and requiring extensive upper-abdominal surgery should be carefully counseled and evaluated before performing cytoreductive surgery. Patients sustaining intraoperative upper-abdominal visceral injuries had the highest risk of developing both severe postoperative complications and TTC >42 days. These patients should be carefully monitored in the postoperative period, to promptly manage complications, speed recovery, and reduce delays in the initiation of adjuvant chemotherapy. In our analysis, patient performance status was the only preoperative modifiable risk factor for TTC >42 days. Further research should be conducted to evaluate the benefit of multimodality pre-habilitation programs as adjuncts to the ERAS protocol, to assess whether such programs could provide incremental improvements in postoperative complication rates, time to chemotherapy, and long-term survival.

## Figures and Tables

**Figure 1 cancers-14-04181-f001:**
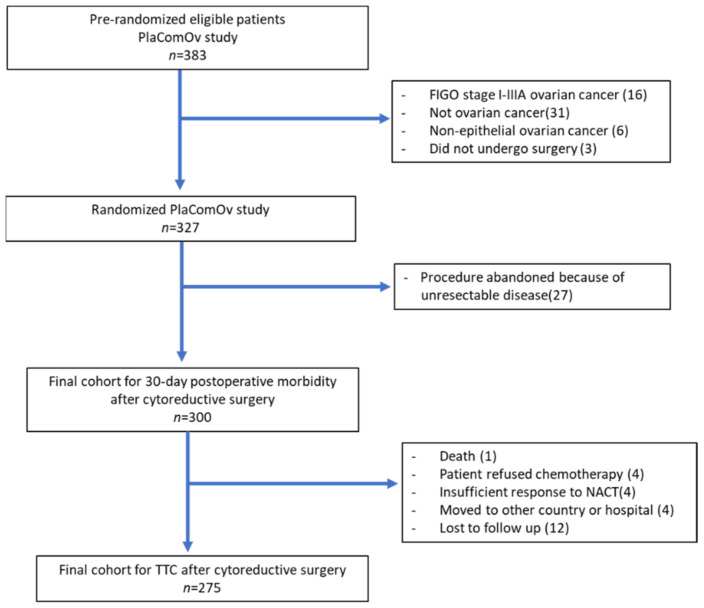
Study flow diagram (FIGO: International Federation of Obstetrics and Gynecology).

**Figure 2 cancers-14-04181-f002:**
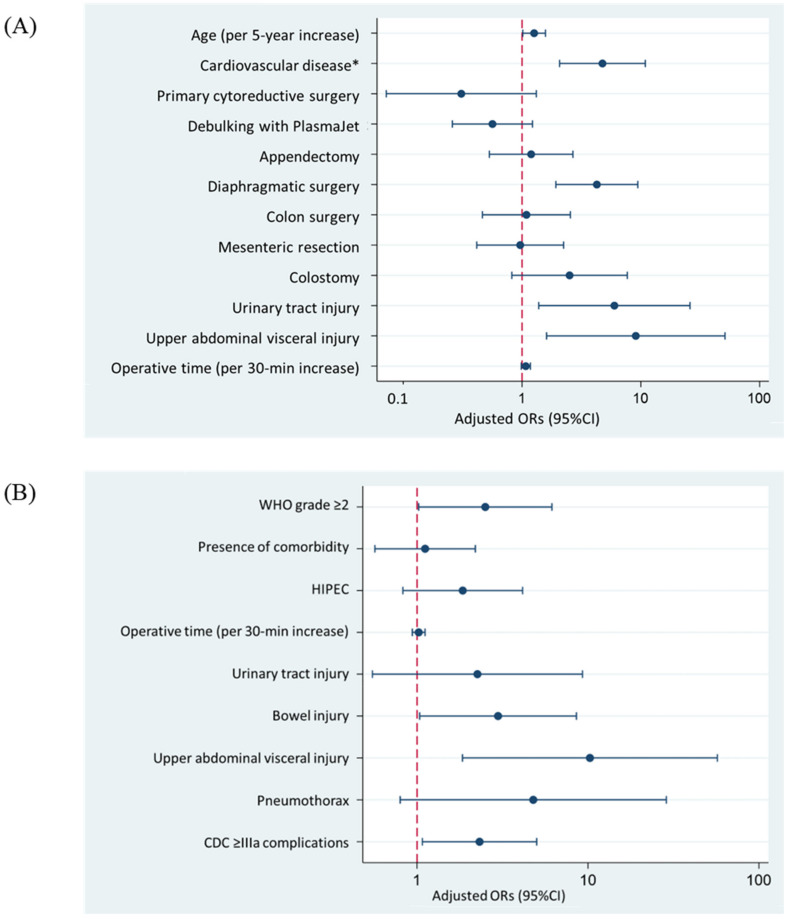
(**A**) Factors related to Clavien–Dindo classification ≥IIIa complications. (**B**) Factors related to time to initiating adjuvant chemotherapy >42 days. * Cardiovascular disease was defined as any of the following: myocardial infarction, stroke, and peripheral vascular disease. WHO: World Health Organization, HIPEC: hyperthermic intraperitoneal chemotherapy, CDC: Clavien–Dindo classification.

**Table 1 cancers-14-04181-t001:** Baseline characteristics (*N* = 300).

Variable	Value (%)
Age, mean ± SD (year)	65 ± 10.4
BMI, mean ± SD (kg/m^2^)	25 ± 4.7
CA125 at diagnosis, median (Q1–Q3) (kU/L)	807 (301–2077)
Daily smoker	29 (9.7)
Daily alcohol drinking	82 (27.4)
WHO performance status	
0: able to all normal activity	175 (58.3)
1: able to carry out light work	105 (35.0)
2: capable of self-care but not any work	12 (4.0)
3: capable of limited self-care, confined to bed or chair >50% of waking hours	8 (2.7)
4: disabled	0 (0)
Comorbidity	
Diabetes mellitus	27 (9.0)
Hypertension	79 (26.3)
Cardiovascular disease *	54 (18.0)
Primary cytoreductive surgery	45 (15.0)
Extensive surgery **	209 (69.7)
HIPEC procedure	61 (20.3)
Debulking with PlasmaJet	138 (46)
Surgical outcome	
Complete cytoreduction (no visible tumor)	235 (78.3)
Optimal cytoreduction (residual tumor ≤1 cm)	50 (16.7)
Suboptimal cytoreduction (residual tumor >1 cm)	15 (5.0)
FIGO stage	
Stage IIIB	21 (7.0)
Stage IIIC	187 (62.3)
Stage IV	92 (30.7)
Post-surgery admission	
Nursing ward	174 (58.0)
Post anesthetic care unit	53 (17.7)
Intensive care unit	73 (24.3)
Histology	
Serous	287 (95.7)
Mucinous	2 (0.7)
Endometrioid	4 (1.3)
Clear cell	6 (2.0)
Mixed epithelial carcinoma	1 (0.3)

BMI: body mass index, WHO: World Health Organization, CA125: cancer antigen 125, Q: interquartile, HIPEC: hyperthermic intraperitoneal chemotherapy FIGO: International Federation of Obstetrics and Gynecology. * Cardiovascular disease was defined as any of the following: myocardial infarction, stroke, and peripheral vascular disease. ** Extensive surgery was defined as any of the following procedures: peritonectomy, diaphragmatic peritonectomy, resection of subcapsular liver metastases, splenectomy, and bowel resection or resection of extra-abdominal metastatic sites.

**Table 2 cancers-14-04181-t002:** Type and frequency of 30-day Clavien–Dindo grade ≥IIIa complications (*N* = 300).

Type	Clavien–Dindo Grade	*N* (%)
Surgical Complications		
Intraabdominal bleeding	IIIb	1 (0.3)
Re-exploration for peritonitis	IIIb	4 (1.3)
Anastomosis leakage	IIIb	2 (0.7)
Gastric leakage	IIIb	1 (0.3)
Pancreatic leakage	IIIa IIIb	1 (0.3) 1 (0.3)
Intraabdominal abscess	IIIa IIIb	4 (1.3) 1 (0.3)
Intestinal perforation	IIIa	1 (0.3)
Splenic hemorrhage	IIIa	1 (0.3)
Pneumothorax	IIIa	4 (1.3)
Wound dehiscence	IIIa	1 (0.3)
Acute cholecystitis	IIIb	1 (0.3)
Medical complications		
Cardiovascular related complications *	IVa	19 (6.3)
Respiratory insufficiency	IVa	11 (3.7)
Massive pulmonary embolism	IVa	1 (0.3)
Acute kidney injury	IVa	1 (0.3)
Acute transaminitis	IVa	1 (0.3)
30-day mortality	V	1 (0.3)
One patient can endure more than one complication		57 complications/51 patients

* Cardiovascular-related complications were defined as any of the following: myocardial infarction, stroke, and peripheral vascular disease.

**Table 3 cancers-14-04181-t003:** Factors related to Clavien–Dindo classification grade ≥ IIIa complications.

Variables	Univariable Analysis Unadjusted OR (95%CI)	*p*-Value	Multivariable Analysis Adjusted OR (95%CI)	*p*-Value
**Pre-operative factor**				
**Age (per 5-year increase)**	**1.19 (1.01–1.40)**	**0.043**	**1.26 (1.02–1.58)**	**0.036**
BMI (per kg/m^2^ increase)	0.98 (0.92–1.05)	0.57		
WHO performance status ≥2	0.72 (0.27–1.96)	0.52		
Daily smoker	1.62 (0.66–4.06)	0.29		
Diabetes mellitus	1.52 (0.85–2.69)	0.16		
Hypertension	1.10 (0.75–1.63)	0.62		
**Cardiovascular disease ***	**3.67 (1.87–7.19)**	**<0.001**	**4.75 (2.07–10.91)**	**<0.001**
Primary cytoreductive surgery	0.40 (0.14–1.16)	0.09	0.31 (0.07–1.32)	0.11
**Intraoperative procedure**				
Pelvic peritonectomy	1.62 (0.86–3.03)	0.14		
Bladder surgery	1.43 (0.78–2.62)	0.25		
Small bowel surgery	1.22 (0.72–2.08)	0.45		
Colon surgery	1.85 (0.97–3.51)	0.06	1.09 (0.46–2.55)	0.85
Appendectomy	1.85 (0.98–3.51)	0.06	1.19 (0.53–2.68)	0.67
Mesenteric resection	1.92 (1.02–3.62)	0.04	0.97 (0.42–2.24)	0.93
Partial hepatectomy	1.06 (0.44–2.57)	0.89		
Splenectomy	1.51 (0.53–4.32)	0.44		
Pelvic lymph node resection	1.40 (0.60–3.29)	0.43		
Paraaortic lymph node resection	1.51 (0.53–4.32)	0.44		
**Diaphragmatic surgery**	**4.01 (2.12–7.57)**	**<0.001**	**4.26 (1.93–9.43)**	**<0.001**
Colostomy	2.82 (1.23–6.51)	0.02	2.51 (0.82–7.69)	0.11
HIPEC procedure	1.13 (0.54–2.37)	0.74		
Debulking with PlasmaJet	0.59 (0.32–1.09)	0.09	0.56 (0.26–1.22)	0.15
Operative time (per 30-min increase)	1.10 (1.02–1.85)	0.012	1.08 (0.98–1.18)	0.11
**Intraoperative injury**				
**Urinary tract injury**	**3.71 (1.13–12.02)**	**0.031**	**6.00 (1.38–25.93)**	**0.017**
Bowel injury	2.05 (0.76–5.58)	0.16		
**Upper-abdominal visceral injury ****	**5.24 (1.46–18.83)**	**0.011**	**9.07 (1.61–51.08)**	**0.012**
Pneumothorax	3.01 (0.69–13.03)	0.14		
Blood loss >1 L	1.65 (0.89–3.09)	0.11		
**Postoperative factor**				
Complete cytoreduction	1.34 (0.62–2.93)	0.46		
FIGO stage				
Stage IIIB (reference)				
Stage IIIC	1.31 (0.38–4.48)	0.66		
Stage IV	1.19 (0.60–2.36)	0.61		

BMI: Body mass index, WHO: world health organization, HIPEC: hyperthermic intraperitoneal chemotherapy FIGO: International federation of obstetrics and gynecology, * cardiovascular disease was defined as any of the following disease: myocardial infarction, stroke, peripheral vascular disease, ** Upper abdominal visceral injury: pancreas, stomach, liver or spleen. A bold font denotes factors that are significant in both a univariate and multivariate model.

**Table 4 cancers-14-04181-t004:** Factors related to time to initiating chemotherapy >42 days.

Variables	Univariable Analysis Unadjusted OR (95% CI)	*p*-Value	Multivariable Analysis Adjusted OR (95% CI)	*p*-Value
**Preoperative factor**				
Age (per 5-year increase)	1.02 (0.89–1.18)	0.75		
BMI (per kg/m^2^ increase)	1.00 (0.96–1.07)	0.70		
**WHO performance status ≥2**	**1.89 (0.89–4.01)**	**0.09**	**2.51 (1.02–6.15)**	**0.045**
Daily smoker	1.66 (0.69–3.95)	0.25		
Presence of comorbidity *	1.67 (0.92–3.04)	0.09	1.12 (0.57–2.19)	0.75
Primary cytoreductive surgery	1.08 (0.51–2.30)	0.84		
**Intraoperative factor**				
Extensive surgery **	0.87 (0.48–1.56)	0.63		
Debulking with PlasmaJet	0.77 (0.45–1.32)	0.34		
HIPEC procedure	2.05 (1.09–3.83)	0.03	1.85 (0.83–4.15)	0.13
Operative time (per 30-min increase)	1.06 (0.98–1.13)	0.09	1.02 (0.94–1.11)	0.52
**Intraoperative injury**				
Urinary tract injury	2.96 (0.83–10.53)	0.09	2.26 (0.55–9.29)	0.26
**Bowel injury**	**3.47 (1.35–8.94)**	**0.01**	**2.98 (1.04–8.56)**	**0.043**
**Upper abdominal visceral injury *****	**6.06 (1.47–24.91)**	**0.01**	**10.26 (1.85–57.13)**	**0.008**
Pneumothorax	7.50 (1.42–39.56)	0.02	4.79 (0.79–28.75)	0.08
Blood loss >1 L	1.09 (0.624–1.910)	0.78		
**Postoperative factor**				
Complete cytoreduction	0.74 (0.390–1.425)	0.38		
FIGO stage				
Stage IIIB (reference)				
Stage IIIC	1.16 (0.34–4.01)	0.82		
Stage IV	1.36 (0.74–2.50)	0.32		
**Post-operative complications CDC ≥IIIa**	**3.13 (1.63–6.02)**	**0.001**	**2.32 (1.08–5.01)**	**0.032**

BMI: Body mass index, WHO: world health organization, FIGO: International federation of obstetrics and gynecology, HIPEC: hyperthermic intraperitoneal chemotherapy, CDC: Clavien-Dindo classification,* Presence of comorbidity: ≥1 of the following comorbidity: diabetes mellitus, hypertension, cardiac disease,** Extensive surgery was defined as any of the following procedures: peritonectomy, diaphragmatic peritonectomy, resection of subcapsular liver metastases, splenectomy, bowel resection or resection of extra-abdominal metastatic sites. *** Upper abdominal visceral injury: any injury of pancreas, stomach, liver or spleen. A bold font denotes factors that are significant in both a univariate and multivariate model.

## Data Availability

The data underlying this article will be shared on a reasonable request to the corresponding author.

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
