# Peer review of "Factors Predicting 30-Day Grade IIIa–V Clavien–Dindo Classification Complications and Delayed Chemotherapy Initiation after Cytoreductive Surgery for Advanced-Stage Ovarian Cancer: A Prospective Cohort Study"

_cancers, 2022, doi:10.3390/cancers14174181_

Round 1

Reviewer 1 Report

In patients treated with primary debulking surgery (PDS), delaying the start of chemotherapy by more than 28 days may have an adverse effect on progression-free survival. This association has been confirmed by some relevant studies, such as Hofstetter et al. (2013; PMID: 23877013) or Liu et al (2018; PMID: 30519317), both unfortunately not cited in the submitted manuscript. Nonetheless, the authors made a commendable effort to perform a post hoc analysis of a multicenter prospective study evaluating the impact of major postoperative complications (Clavien-Dindo classification > IIIa) delaying the start of chemotherapy. The authors chose 42 days (6 weeks) as the cut-off point for delayed initiation of chemotherapy, which is supported by three citations.

Unfortunately, the study design and presentation of results is somewhat misleading and should be corrected before final acceptance for publication. A major concern is that the composition of the cohort studied is remarkable: only 15% of the patients studied received PDS, while the vast majority of them (85%) received interval debulking surgery (IDS ) after NACT. Justification for such a high rate of patients qualifying for NACT? If only frail patients with inoperable tumors after NACT were included, this fact could have a relevant impact on the perioperative course. Likewise, the obtained NACT itself could also be a confounding factor. Furthermore, the number of chemotherapy cycles and chemotherapy regimens remain elusive to the reader.

Also, the exact number and percentage of each procedure (similar to the nature and frequency of each type of complication) should be reported, instead of the global category of “extended surgery” defined as “any of” the procedures on the list.

Despite these methodological criticisms, I think the results are of clinical importance (although they are mainly applicable to a subset of patients after NACT). The use of an established classification system for postoperative complications (Clavined-Dindo) increases the potential value of this publication.

I see two solutions: a) either limit the study group to patients treated with IDS after NACT and resubmit a paper pertaining to this well-defined category of patients or b) better justify the actual study design and clearly indicate within the abstract and the manuscript that the results are particularly valid for patients treated with NACT/IDS.

Author Response

Response to reviewer1

In patients treated with primary debulking surgery (PDS), delaying the start of chemotherapy by more than 28 days may have an adverse effect on progression-free survival. This association has been confirmed by some relevant studies, such as Hofstetter et al. (2013; PMID: 23877013) or Liu et al (2018; PMID: 30519317), both unfortunately not cited in the submitted manuscript. Nonetheless, the authors made a commendable effort to perform a post hoc analysis of a multicenter prospective study evaluating the impact of major postoperative complications (Clavien-Dindo classification > IIIa) delaying the start of chemotherapy. The authors chose 42 days (6 weeks) as the cut-off point for delayed initiation of chemotherapy, which is supported by three citations.

Response:

Thank you for your thorough review and suggestions. We agree that patients' survival outcomes can be affected by the increased time to initiate adjuvant chemotherapy. There are several cut-off values used as references for delayed chemotherapy initiation. It is, however, unclear whether there is an international consensus on this topic. The cut-off we used in our study, of more than 42 days after surgery, is based on the study of the Dutch cancer registry (n=4097) by Timmermans et al., PMID: 30001834. We appreciate you provided the additional references, and have included them in the introduction as well as the discussion.

Paragraph 2

Unfortunately, the study design and presentation of results is somewhat misleading and should be corrected before final acceptance for publication. A major concern is that the composition of the cohort studied is remarkable: only 15% of the patients studied received PDS, while the vast majority of them (85%) received interval debulking surgery (IDS) after NACT. Justification for such a high rate of patients qualifying for NACT? If only frail patients with inoperable tumors after NACT were included, this fact could have a relevant impact on the perioperative course. Likewise, the obtained NACT itself could also be a confounding factor. Furthermore, the number of chemotherapy cycles and chemotherapy regimens remain elusive to the reader.

 Response:

Thank you for your feedback on this issue. As part of our study, we aimed to evaluate factors associated with Clavien-Dindo classification (CDC) ≥ IIIa complications and delays in initiation of chemotherapy (TTC >42 days) in advanced stage ovarian cancer patients who underwent either primary cytoreductive surgery (PCS) or interval cytoreductive surgery (ICS).

In the Netherlands, ICS is often decided in case of advanced ovarian cancer. This is supported by the publication of the results of the prospective EORTC 55971 trial by Vergote et al., in 2010 (PMID: 20818904). This study showed that neoadjuvant chemotherapy followed by interval cytoreductive surgery was not inferior to primary cytoreductive surgery followed by chemotherapy as a treatment option for patients with bulky stage IIIC or IV ovarian carcinoma. Complete resection of all macroscopic disease, whether performed as primary treatment or after neoadjuvant chemotherapy, remains the objective whenever cytoreductive surgery is performed. Results of this study were confirmed in the pooled analyses of individual patient data from the EORTC 55971 and CHORUS trials. (PMID: 30413383). As a result, in advanced ovarian cancer ICS increased from 28% in 2004 to 71% in 2013 in the Netherlands. (Eggink et al., PMID: 27090157).

In our study, type of surgery (PCS vs ICS) was not independently associated with CDC grade ≥IIIa or TTC >42 days. According to your suggestion, we have conducted a sensitivity analysis (lines 271-281)

Consistent with the main analysis, a sensitivity analysis of patients who received ICS (n=255), demonstrated four factors were independently related to CDC grade ≥ IIIa: cardiovascular disease (aORs 4.01, 95%CI 1.66- 9.68, p=0.002), diaphragmatic surgery (aORs 3.90, 95%CI 1.74-8.70, p<0.001), intra-operative urinary tract injury (aORs 6.90, 95%CI 1.50-31.69, p=0.013), and upper abdominal visceral injury (aORs 9.97, 95%CI 1.53-65.08, p=0.016) (Table S5). Increasing age was no longer an independent predictor, but the effect size was similar in magnitude to that observed in the model including all patients, and the 95%CI were consistent with an increase risk.

Patients who experienced intra-operative bowel injury (aORs 3.27, 95%CI 1.03-10.37, p=0.045), upper abdominal visceral injury (aORs 17.59, 95%CI 1.87-165.36, p=0.012), and post-operative CDC grade ≥ IIIa complications (aORs 2.47, 95%CI 1.11-5.49, p=0.027) had a significantly higher adjusted odds of developing TTC >42 days (TableS6). These results are similar to the entire cohort analysis, except that WHO performance status ≥2 did not meet criteria for inclusion in the multivariable sensitivity analysis although the effect size and the 95% CI were consistent with an increase risk in the univariable analysis

Paragraph 3-5

              Also, the exact number and percentage of each procedure (similar to the nature and frequency of each type of complication) should be reported, instead of the global category of “extended surgery” defined as “any of” the procedures on the list.

Despite these methodological criticisms, I think the results are of clinical importance (although they are mainly applicable to a subset of patients after NACT). The use of an established classification system for postoperative complications (Clavien-Dindo) increases the potential value of this publication.

I see two solutions: a) either limit the study group to patients treated with IDS after NACT and resubmit a paper pertaining to this well-defined category of patients or b) better justify the actual study design and clearly indicate within the abstract and the manuscript that the results are particularly valid for patients treated with NACT/IDS.

Response:

We have added the type and frequency of surgical procedure in Table S3. Information on the study population and type of cytoreductive surgery are now clarified in the abstract line 41 and manuscript (line 191-194). The sensitivity analysis on ICS patients has been added which provides additional outcome information on the study population according to your comment (Table S5 and S6)

Reviewer 2 Report

Manuscript details
Journal: Cancers

Manuscript ID: cancers-1816681
Type of manuscript: Article
Title: Factors predicting 30-day Grade IIIa-V Clavien-Dindo Classification Complications and Delayed Chemotherapy Initiation after Cytoreductive Surgery for Advanced Stage Ovarian Cancer: A Prospective Cohort Study
Authors: Malika Kengsakul *, Gatske M. Nieuwenhuyzen-de Boer, Suwasin Udomkarnjananun, Stephen J. Kerr, Helena C. Van Doorn, Heleen J. Van Beekhuizen 

Comment

A large prospective cohort study to evaluate the factors predicting 343 both severe post-operative complications and time to adjuvant chemotherapy >42 days was performed in this study.

Work primary aim was to analyze factors associated with severe 30-day post-operative morbidity, defined as Clavien-Dindo (CDC) classification grade ≥IIIa, in patients who underwent either primary or interval cytoreductive surgery for advanced stage epithelial ovarian cancer.

Secondary aim was to evaluate factors associated with a delay in initiating chemotherapy.

Identification of these potential risk factors and their consequences would offer an opportunity to improve perioperative care in the future.

This study is a nice piece of work and provide useful suggestion to manage postoperative complications in patients with advanced epithelial ovarian cancer.

Suggestion

Taking into consideration the results reported into the manuscript, it is suggested that it should be submitted to a specialized journal dealing with Gynaecological Oncology. 

Author Response

Response to reviewer 2

A large prospective cohort study to evaluate the factors predicting 343 both severe post-operative complications and time to adjuvant chemotherapy >42 days was performed in this study. Work primary aim was to analyze factors associated with severe 30-day post-operative morbidity, defined as Clavien-Dindo (CDC) classification grade ≥IIIa, in patients who underwent either primary or interval cytoreductive surgery for advanced stage epithelial ovarian cancer. Secondary aim was to evaluate factors associated with a delay in initiating chemotherapy. Identification of these potential risk factors and their consequences would offer an opportunity to improve perioperative care in the future.

This study is a nice piece of work and provide useful suggestion to manage postoperative complications in patients with advanced epithelial ovarian cancer.

Suggestion

Taking into consideration the results reported into the manuscript, it is suggested that it should be submitted to a specialized journal dealing with Gynaecological Oncology. 

Response:

Thank you for your positive feedback on our manuscript. We want to publish in Cancers, as a highly respected oncology journal. We believe that our works will be widely distributed to the intended readers if our article is accepted

Reviewer 3 Report

In this report the authors assess pre and postoperative factors associated with delayed initiation of chemotherapy, known to result in diminished outcomes.  Strengths include the prospective data collection, the large sample size, and the use of the CDC system to classify outcomes.

Weaknesses include the fact that half of the patients were randomized to the use of the Plasma Jet system.  While this was not significantly associated with morbidity, it is unclear whether the surgeons were familiar with this instrument or whether this may have affected patient selection or intraoperative factors.  Additional discussion about the parent trial would be helpful, and an understanding whether this analysis of surgical outcomes was a pre-planned component of the study. 

Additional comments include the fact that despite a majority of patients undergoing neoadjuvant chemo, there seems to be a high rate of ultraradical procedures performed.  The number of chemo regimens received prior to surgery is not considered as a variable.

Ultimately the report adds to existing information about the impact of surgical decision-making on disease outcomes but the distribution of surgical interventions (majority NCAT, 20% HIPEC) and the potential confounding effect of the PlasmaJet analysis make it unclear how generalizable these results are.

Author Response

Response to reviewer 3

In this report the authors assess pre and postoperative factors associated with delayed initiation of chemotherapy, known to result in diminished outcomes.  Strengths include the prospective data collection, the large sample size, and the use of the CDC system to classify outcomes.

Weaknesses include the fact that half of the patients were randomized to the use of the Plasma Jet system.  While this was not significantly associated with morbidity, it is unclear whether the surgeons were familiar with this instrument or whether this may have affected patient selection or intraoperative factors.  Additional discussion about the parent trial would be helpful, and an understanding whether this analysis of surgical outcomes was a pre-planned component of the study. 

Additional comments include the fact that despite a majority of patients undergoing neoadjuvant chemo, there seems to be a high rate of ultraradical procedures performed.  The number of chemo regimens received prior to surgery is not considered as a variable.

Ultimately the report adds to existing information about the impact of surgical decision-making on disease outcomes but the distribution of surgical interventions (majority NCAT, 20% HIPEC) and the potential confounding effect of the PlasmaJet analysis make it unclear how generalizable these results are.

Response:

Thank you very much for your comment on the generalizable of the result. We have added the information about the parent study (PlaComOv study) and NACT-ICS protocol in methods (line 107-122) as well as in the discussion (line 284-289). Prior to the randomization of PlaComOv study, gynecological oncologist involved in the study were trained and certified in the use of PlasmaJet device. Since 2012, all ovarian cancer surgery in the Netherlands is performed by well-trained gynecological oncologist at registered cancer centers, thus increasing the rate of ICS, complete cytoreduction and extensive procedures (Eggink et al., PMID: 27090157). In the PlaComOv study, the rate of surgical complications was similar between patients receiving standard care plus PlasmaJet, versus standard care alone (Nieuwenhuyzen-de Boer et al., PMID: 35552938). In the present study, moment of surgery (PCS versus ICS), adding HIPEC and the use of PlasmaJet device did not show an independent association with CDC grade≥ IIIa or TTC >42 days. We believed that our results are therefore generalizable and could be useful to guide clinicians who care for ovarian cancer patients.

Reviewer 4 Report

I wish to thank you for giving the opportunity to review this interesting manuscript.

The study presents a cohort of 300 patients with advanced ovarian cancer treated with cytoreductive surgery regarding risk factors for severe complications and delayed chemotherapy initiation. The manuscript is well written, and the methodology is clearly defined.

I have several remarks:

1.       Several studies have shown other parameters which are correlated with surgical outcomes such as ASA score, Hypoalbuminemia (Albumin<3.5) and PCI (Peritoneal cancer index). Were these parameters assessed, if not why there are not included in the analysis?

2.       The parameters mentioned in comment number 1 were also reported to correlates with TCC delay. Were these parameters assessed?

3.       Regarding table 2. (Type and frequency of 30-day CDC grade ≥IIIa complications), I would recommend adding number of patients that had complication grade IV.

4.       In Table 1 cardiovascular disease was defined as any of the following: myocardial infarction, stroke, peripheral vascular. In Table 2 one patient had CVA, is he included in the cardiovascular related complications also? Why CVA is separated medical complication?

5.       I would recommend elaborating more on limitations (discussion section): patients with low performance status were excluded which would make possible selection bias, some risk factors weren’t assessed (see comment number 1 and 2).

Author Response

Response to reviewer 4

I wish to thank you for giving the opportunity to review this interesting manuscript. The study presents a cohort of 300 patients with advanced ovarian cancer treated with cytoreductive surgery regarding risk factors for severe complications and delayed chemotherapy initiation. The manuscript is well written, and the methodology is clearly defined.

I have several remarks:

  1. Several studies have shown other parameters which are correlated with surgical outcomes such as ASA score, Hypoalbuminemia (Albumin<3.5) and PCI (Peritoneal cancer index). Were these parameters assessed, if not why there are not included in the analysis?

  1. The parameters mentioned in comment number 1 were also reported to correlates with TCC delay. Were these parameters assessed?

Response

Thank you for your comprehensive review and suggestion. We have revised our manuscript as follows:

Response to remark1 and 2

We agree that the parameters mentioned by the reviewer are important and interesting factors. Both ASA and pre-operative albumin level were not routinely recorded in our study, and although tumor distribution pattern was described during operation, PCI was not introduced into our routine clinical practice until 2019. Regrettably therefore, these latter three factors were not available for analysis in this study. Since WHO performance status and ASA score have been shown to perform similarly in risk adjustment models predicting outcomes after cancer surgery (Young et al., PMID: 24996034), we think that using the WHO performance status overcomes the issue about missing ASA status.

  1. Regarding table 2. (Type and frequency of 30-day CDC grade ≥IIIa complications), I would recommend adding number of patients that had complication grade IV.

Response to remark 3

We have added number of grade IV complication in Table 2 according to your suggestion.

  1. In Table 1 cardiovascular disease was defined as any of the following: myocardial infarction, stroke, peripheral vascular. In Table 2 one patient had CVA, is he included in the cardiovascular related complications also? Why CVA is separated medical complication?

Response to remark 4:

Thank you for this suggestion. We have revised Table 2 to include post-operative cerebrovascular accidents in the cardiovascular related complications.

  1. I would recommend elaborating more on limitations (discussion section): patients with low performance status were excluded which would make possible selection bias, some risk factors weren’t assessed (see comment number 1 and 2).

Response to remark5

We agree with your suggestion. We have added information in strength and limitation in line 399-411 as follows:

Although data from this study was from an RCT, we excluded patients who declined surgery for deterioration in performance status, patients with progressive disease and patients in whom surgery was abandoned. In addition, some variables, such as preoperative frailty index, albumin level, intraoperative peritoneal carcinoma index and ERAS compliance which are known to influence outcome were not available in the study database and therefore not included in our analysis. In an attempt to mitigate bias caused by these issues, we adjusted for all potential confounders in our multivariable model, but the possibility of unobserved confounding cannot be excluded.

Round 2

Reviewer 1 Report

The authors made efforts to improve the manuscript, and indeed the work benefited from the improvements. Some concerns remain (particularly the high proportion of NACT/IDS) but should not affect the decision to publish the present work.